# Use of Intravenous Immunoglobulins in Sepsis Therapy—A Clinical View

**DOI:** 10.3390/ijms21155543

**Published:** 2020-08-03

**Authors:** Dominik Jarczak, Stefan Kluge, Axel Nierhaus

**Affiliations:** Department of Intensive Care Medicine, University Medical Center Hamburg-Eppendorf, 20246 Hamburg, Germany; d.jarczak@uke.de (D.J.); s.kluge@uke.de (S.K.)

**Keywords:** immune response, immunoglobulins, sepsis, septic shock, IVIg, IgGAM

## Abstract

Sepsis is a life-threatening organ dysfunction, defined by a dysregulated host immune response to infection. During sepsis, the finely tuned system of immunity, inflammation and anti-inflammation is disturbed in a variety of ways. Both pro-inflammatory and anti-inflammatory pathways are upregulated, activation of the coagulation cascade and complement and sepsis-induced lymphopenia occur. Due to the manifold interactions in this network, the use of IgM-enriched intravenous immunoglobulins seems to be a promising therapeutic approach. Unfortunately, there is still a lack of evidence-based data to answer the important questions of appropriate patient populations, optimal timing and dosage of intravenous immunoglobulins. With this review, we aim to provide an overview of the role of immunoglobulins, with emphasis on IgM-enriched formulations, in the therapy of adult patients with sepsis and septic shock.

## 1. Introduction

Sepsis is a life-threatening, dysregulated immune response that occurs, when the body’s defensive reactions against infection damage its own tissues and organs [1]. In 2017, an estimated 48.9 million cases of sepsis were recorded worldwide with 11.0 million sepsis-related deaths, representing 19.7% of all global deaths [2]. Due to continuous progress in the understanding of the underlying pathology and immunological mechanisms, the definition of sepsis as a clinical syndrome is subject to constant development. The current consensus definition (“Sepsis-3”) emphasizes for the first time the crucial role of the innate and adaptive immune response in the development of the clinical syndrome. Despite the enormous efforts made during the last three decades of clinical and experimental research, the available therapeutic armamentarium to positively affect the course of the disease remains restricted. Even today, the mortality of septic shock, the most severe subgroup of sepsis, lies in the range of more than 50% in North America and Europe [3].

## 2. The Changing Immune System in Sepsis

In contrast to an uncomplicated and often localized infection, sepsis leads to a system-wide release of cytokines, mediators and pathogen-related molecules (“cytokine storm”) [4]. The starting signal for the activation of numerous signal cascades is given by the recognition of pathogen-derived molecules (pathogen-associated molecular patterns, PAMPs—e.g., endo- and exotoxins, lipids, DNA) or endogenous host-derived danger signals (damage-associated molecular patterns, DAMPs) via specific receptors (toll-like receptors, TLR) on the surface of monocytes and antigen-presenting cells (APCs). This initiates the clinical syndrome of sepsis through expression of genes involved in inflammation, cellular metabolism and adaptive immunity [5].

Pro- and anti-inflammatory pathways are upregulated, leading to inflammation and progressive tissue damage, ultimately causing multi-organ dysfunction. Simultaneous immunosuppression by downregulation of activating cell surface molecules, T cell exhaustion and increased apoptosis of immune cells invariably accounts for “immunoparalysis” during the later stages of the disease, making the affected patients susceptible for nosocomial infections, viral reactivation and opportunistic pathogens (Figure 1) [6,7].

### 2.1. Early Pro- and Anti-Inflammatory Responses

Binding of PAMPs and DAMPs to TLRs on monocytes and APCs causes signal transduction and induces the translocation of nuclear factor kappa-light-chain-enhancer of activated B cells (NF-κB) to the nucleus. In consequence, so-called “early activation genes” are expressed, including various pro-inflammatory interleukins (IL), e.g., IL-1, IL-12, IL-18, tumor necrosis factor alpha (TNF-α) and interferons (IFNs). These changes lead to activation of further cascades of inflammatory cytokines (e.g., IL-6, IL-8, IFN-γ), the coagulation cascade and complement in addition to a downregulation of adaptive immunity components [8]. As a result, increased levels of both pro-inflammatory and anti-inflammatory cytokines can be detected in the early stages of sepsis [7,9,10].

### 2.2. Late and Persistent Immunosuppressive Events

Even though the early systemic inflammatory response is generally considered the hallmark of sepsis, there is also a significant component of immunosuppression that occurs both early and late in the host sepsis response [8]. The role of B-lymphocytes in sepsis exceeds the production and secretion of immunoglobulins, they also modulate the innate immune response, produce cytokines and act as APCs [11,12]. In the early course of sepsis, a decrease in lymphocytes, monocytes and antigen-presenting dendritic cells has been observed [13,14]. The delicate mechanisms underlying sepsis-induced lymphopenia have not yet been conclusively explained. As a possible explanation, recruitment of lymphocytes from the peripheral circulation into areas of inflammation and infection is discussed, but most data suggest that apoptosis causes sepsis-induced lymphopenia [15,16,17]. The persistence of lymphopenia and therefore also the lower levels of immunoglobulins over the course of sepsis is closely associated with increased mortality [17,18].

### 2.3. Monocytes and Antigen-Presenting Cells

In addition to sepsis-induced lymphopenia, an increase in apoptosis of monocytes and APCs could be shown, which is accompanied by a significant loss of pro-inflammatory cytokine production [13,14,19,20,21,22,23,24,25,26,27]. The remaining cells also present with a decreased expression of the human leukocyte antigen DR (HLA-DR) on their surface, resulting in a diminished ability to recognize pathogens and to interact with T cell receptors via opsonisation. As a consequence, this leads to a lack of Th1- and Th2-response as a component of the adaptive immune response [28]. The inability of monocytes to restore normal expression levels of HLA-DR has been shown to predict poor outcome in sepsis [29,30]. Furthermore, it has also been shown that endotoxin tolerance in the early course of sepsis is also associated with further clinical deterioration [31].

### 2.4. Neutrophils

As part of the innate immune system, neutrophils belong to the first line of defence against pathogens. Severe bacterial infections result in emergency granulocyte formation and the release of both mature and immature forms of neutrophils from the bone marrow. When activated via PAMPs and DAMPs in septic patients, these cells show reduced phagocytosis and oxidative burst capacity [32]. High levels of immature granulocytes in sepsis are associated with clinical deterioration, since these have shown increased spontaneous production and liberation of neutrophil extracellular traps (NETs) [33,34]. NETs are diffuse extracellular structures of decondensed chromatin with granular and nuclear proteins [35,36]. These structures have the potential to capture a wide range of pathogens, including Gram-positive and Gram-negative bacteria, viruses, yeasts, but also protozoa and parasites that cannot be phagocytized due to their size [37]. It is known that cytokines, chemokines, platelet agonists and antibodies can also trigger their release. The pronounced presence of NETs in tissues or vessels due to insufficient removal or overproduction is linked to hypercoagulation and endothelial injury [34,38,39].

### 2.5. Myeloid-Derived Suppressor Cells

During granulopoiesis as a response to acute infection, immature myeloid cells can shift into the peripheral blood and become functionally active. These myeloid-derived suppressor cells (MDSCs) release various anti-inflammatory cytokines (e.g., IL-10 and transforming growth factor β, TGF-β), leading to further immunosuppression [8,40]. These immunosuppressing properties have been the focus of extensive research in the context of malignant diseases, but the knowledge in relation to sepsis is still scarce [41].

### 2.6. B-Lymphocytes

Although many details are known about the function of B-lymphocytes in the context of sepsis, their clinical relevance cannot yet be conclusively determined. For an effective host protection, B cell function and antibody expression are decisive factors. The production of antibodies is one of the crucial functions of B cells after differentiation into high-affinity antibody-secreting plasma cells [42]. Within the framework of the adaptive immune response, interaction with dendritic cells, macrophages, T- and other B- cells leads to clonal expansion and finally to the production of specific antibodies. However, in the early phase of sepsis, B cells can also be activated by pathogens themselves via pathogen recognition receptors (PRRs), which leads to an initial immune response by innate like B cells [11,43,44].

Just at the beginning of sepsis, the number of B-lymphocytes in the peripheral blood is often reduced [45,46]. In addition to increased apoptosis, increased migration from the circulation into the tissue or reduced production of these B cells in favour of the production of monocytes and neutrophils as part of an emergency haematopoiesis is possible as a cause. A recent study by Dong et al. showed that in septic shock, a severe functional impairment of B cells is present in non-survivors, resulting in both lower serum immunoglobulin M (IgM) levels and in lower IgM production upon B cell stimulation. Examining five subsets of peripheral blood B cells (immature/transitional B cells, naive B cells, tissue-like memory B cells, resting memory B cells, and activated memory B cells), the authors demonstrated a distinct redistribution of these subsets in septic shock patients when compared to healthy controls [47].

A meta-analysis could show that the number of circulating B cells is significantly higher in sepsis survivors than in sepsis non-survivors, especially within the first 24 h after the onset of sepsis [48]. The mechanism presumably responsible for this protective effect seems to be the release of natural antibody IgM, which plays a key role in the fight against gram-negative infections in particular [11]. Furthermore, it could be shown that IgM plasma levels in survivors of sepsis or septic shock were higher in the first 24 h than in non-survivors, which supports the hypothesis of a protective function of B cells through the production of IgM [48]. Interestingly, a similar relationship was also shown in non-septic critically ill patients [49]. However, current data are not yet sufficient to use the measurement of B cells or IgM levels in early sepsis as a prognostic factor for outcome.

### 2.7. Immunoglobulins

Immunoglobulins are glycoproteins secreted by differentiated B cells, so called plasma cells. Each immunoglobulin molecule monomer consists of identical light and heavy chain pairs, held together by disulphide bonds and electrostatic forces. Based on the heavy chain, there are five isotypes of immunoglobulins, IgA, IgD, IgE, IgG and IgM, respectively [50]. The variable regions of immunoglobulins enable non-covalent cross-linking to bacterial and other antigens, whereby the constant region transduces signals in response to antigen-binding. The most important classes within the human humoral immune system are IgA, IgG and IgM. IgA has two subclasses (IgA1, IgA2) and the main function is mucosal immunity. IgG has four subclasses (IgG1–IgG4) and the key functions besides secondary antibody responses are opsonisation and complement activation. The main functions of IgM are complement activation and primary antibody responses. Two types (kappa and lambda) of light chains are also present in the circulation independent of whole immunoglobulin molecules, referred to as free light chains (FLC), which can be detected at abnormal high levels in adult patients with sepsis [51].

## 3. Considerations for the Therapeutic Use of Immunoglobulins

The concept that acquired immunosuppression is a significant event in sepsis and septic shock leads to the hypothesis that stimulation of the immune response and/or substitution of individual immune system components could be a promising therapeutic approach. Within the intricate web of interacting and regulating factors of the immune system and the inflammatory response, polyvalent intravenous immunoglobulins (IVIg) might be a tool for modulating both pro- and anti-inflammatory processes (Figure 2).

The clinical rationale for IVIg therapy in sepsis can be categorized as follows: the role of immunoglobulins in (i) recognition and clearance of pathogens and toxins, (ii) scavenging and inhibition of up- and downstream mediator gene transcription, and (iii) anti-apoptotic effects on immune cells.

Recognition of PAMPs is based on naturally occurring antibodies, which can also act as innate immune receptors. IgG and the complement proteins are the most important opsonins for bacterial clearance. The activation of the classical pathway of the complement system is activated by the interaction of C1 complex with immunoglobulins, acute phase proteins and non-specific activators [52]. Human neutrophils express multiple Fcγ -type cell surface receptors, that are capable of binding IgG, resulting in neutrophil activation via tyrosine kinase pathways and thereby upregulating the expression of phagocytic receptors, that are able to identify and phagocytose pathogens opsonized with IgG and complement proteins [53,54]. Due to IgG deficit, the activation of neutrophils as well as phagocytosis signals may be affected in septic patients, so this population seems to be an optimal cohort for IVIg therapy.

Monocytes and T cells also become activated by superantigen exotoxins released by staphylococci and streptococci. IgG molecules contained in IVIg preparations can inhibit or neutralise these superantigens, thereby preventing superantigen-mediated T cell and monocyte activation [55,56,57].

A key mechanism for regulating host response in inflammatory situations like sepsis or septic shock (as well as other inflammatory diseases) is NF-κB-mediated up-regulation of IL-1 and the IL-1 receptors system [58]. It has been shown, that these components decrease, if IVIg is supplemented in case of hypogammaglobulinaemia or sepsis, as well as IL-1 mediated activity of mononuclear cells in peripheral blood reduces and IL-1 receptor antagonist (IL-1ra) becomes induced [58,59].

In addition to neutralising antibodies in IVIg preparations, naturally occurring auto-antibodies, antiidiotype antibodies and immune proteins in IVIg preparations may also contribute to its immunomodulatory properties [60]. In healthy individuals, autoantibodies neutralising cytokines such as IFN-α, -β and -γ, IL-1α, -2, -4, -6, -8, -10, TNF-α and -β, and soluble TNF receptors have been regularly detected. Therefore, polyclonal IVIg formulations are likely to contain these antibodies, which contribute to cytokine modulation as another important component of anti-inflammatory IVIg activity [61,62,63].

Alas, despite the numerous theoretical advantages clinical studies have shown that the administration of preparations containing only IgG did not lead to a reduction in mortality in patients with sepsis [64,65,66]. However, systematic reviews and meta-analyses suggest that the use of preparations with enriched IgA and IgM (IgGAM) is associated with a higher survival rate [67,68,69,70,71]. A recent meta-analysis, including 19 trials with more than 1500 patients showed a significant reduction in mortality when using IgM- and IgA-enriched immunoglobulins compared to human albumin solution or no special treatment as a control intervention [67]. In addition, a post-hoc analysis of the CIGMA trial demonstrated a significant relative reduction in all-cause mortality of 54–68% using IgM- and IgA enriched immunoglobulins in patients with severe pneumonia and had high C-reactive protein (CRP), low IgM and high CRP/low IgM ratios at baseline compared to placebo [72]. Conversely, patients with normal IgM levels showed a tendency towards higher mortality in the treatment group (10/25 vs. 5/24, *p* = 0.2165), which might suggest that precise identification of the target population is essential. Further, a recent clinical trial, in which patients were treated with either IgGAM or placebo (NaCl), demonstrated a significant decrease in IL-6 and IL-10 levels after 72 h only in the IgGAM group [73]. A positive effect of IgM administration on microvascular perfusion parameters could be demonstrated in humans, confirming previous studies in an animal model of endotoxemia [73,74]. IgM has also been shown to have a positive effect on the integrity of the blood-brain barrier and on septic encephalopathy in rats [75,76].

The decrease of systemic endotoxin levels associated with IgGAM has been shown in animal models. Human studies have focused on neonates and preterm infants, or on neutropenic patients [77,78,79]. Recently, a small controlled study in adult patients with sepsis and septic shock and elevated endotoxin levels demonstrated that IgGAM significantly attenuated LPS levels and had a beneficial effect on sepsis-related coagulopathy in terms of platelet count and fibrinogen concentrations [80].

The precise mechanisms by which IgGAM brings about these effects in patients with sepsis or septic shock are not conclusively clarified until now. In addition to the opsonisation of pathogens and the neutralization of bacterial endotoxins and exotoxins, IgGAM modulates the immune response by attenuating an excessive inflammatory response [80,81,82,83]. The reduced production of IL-2 when IgGAM is used is considered to be one of the proven mechanisms of action [84]. IL-2 (also T cell growth factor, TCGF) is mainly produced by activated T cells and is released after MHC-II-mediated recognition of an antigen. It is mainly autocrine and serves two opposite functions: On the one hand, IL-2 enhances the proliferative response of effector T cells (T_eff_ cells) and natural killer cells (NK cells), on the other hand IL-2 simultaneously controls immune homeostasis by influencing proliferation, differentiation and function of regulatory T cells (T_reg_ cells) [85]. 

Overall, this leads both in vitro and in lectin-stimulated peripheral blood mononuclear cells to a significant inhibition of the alloproliferative response of human T-lymphocytes [84,86]. Immunoglobulin-induced downregulation of IL-2 also causes a lack of activation of B-lymphocytes and decreases the production of pro-inflammatory TNF-α as well as the cytokines of the type 2 T-helper cell response, IL-4 and IL-5 [87].

## 4. Immunoglobulins in Clinical Use

Up to now, the use of intravenous Ig as supportive therapy in sepsis is controversial and not entirely without risk. In some patients, serious adverse reactions consist of the development of a hyperviscosity syndrome with thromboembolic events. Further, acute renal failure has been observed, which was presumably associated with stabilizers contained in the IVIg preparations. IVIg-associated renal failure is most common in patients with pre-existing conditions such as renal impairment, diabetes mellitus, advanced age, volume depletion or concomitant use of other substances known to cause renal toxicity [88]. However, most of these potential complications can be prevented by taking appropriate countermeasures. For example, slow infusion rates and adequate hydration may help to avoid renal failure as well as thromboembolic events [89].

Nevertheless, due to their pleiotropic effects and the potential modulation of both pro- and anti-inflammatory processes, polyvalent intravenous immunoglobulins offer a promising strategy within the context of inflammation and immunity. Experimental studies have shown that polyvalent immunoglobulins can improve phagocytosis of pathogens via opsonisation, neutralize exo- and endotoxins and interact with complement factors, thereby preventing nonspecific activation [80,82,83,84]. Nevertheless, the Surviving Sepsis Campaign (SSC) guidelines suggested against IVIg use in sepsis therapy 2016 due to lack of sufficient evidence of efficacy for preparations containing only IgG [90]. Prepared from a pool of donors, IVIg are widely used in the treatment of haematological, immunological, and neurological diseases. Classic IVIg-preparations contain more than 90% IgG. Since human plasma contains all three immunoglobulin classes, IgGAM preparations are considered physiological [91]. 

Pentaglobin is currently available as a preparation in which the proportion of IgM and IgA is enriched to 12% each, with an IgG proportion of 76%. Pentaglobin also contains toxin-binding and neutralising antibodies against numerous Gram-positive and Gram-negative bacteria and influences the effect of further pro-inflammatory (IFN-γ, IL-6) and anti-inflammatory (IL-10) cytokines during the lymphocyte response [84,92,93]. Another preparation with an even higher proportion of IgM (23%) and IgA (21%) is in clinical development under the working name Trimodulin [72].

Most of the evidence currently available on the use of IgGAM in sepsis or septic shock is based primarily on the use of Pentaglobin in clinical trials. Unfortunately, these data suffer from the fact that, in addition to a large number of different protocols (various dosages, heterogeneous patients), the laboratory parameters analysed are also inconsistent [94]. 

### The Questions of “Who?”, “When?” and “How Much?”

Understanding which patients can generally benefit most from therapy with IgGAM is of high clinical relevance. Previous studies investigating immunomodulatory approaches in the treatment of sepsis often lacked a precise characterization of suitable patients [95,96,97,98,99]. According to the current state of knowledge, the identification of a suitable target population for IgGAM therapy is important. In particular, the idea of “one size fits all” therapy must be critically examined under the current concept of “personalised medicine” and also the lack of cost-effectiveness data [100,101,102,103,104].

In 2005, a meta-analysis investigated 9 trials including 435 patients, and showed an increase of direct ICU costs for the treatment of adult patients with severe sepsis and septic shock with IgGAM by €2037, but this was offset by costs per life saved of €10,565 [105]. This analysis demonstrated also a significantly reduced mortality risk but showed no effect on ICU length of stay. As many other analyses in this context, this was based on small trials of variable quality.

As a complex syndrome, sepsis is characterized by a variety of pathogen- and host-specific factors that show a very dynamic behaviour over time [1]. Especially patients with the appearance of an excessive immune reaction, but also those with the appearance of an immune paralysis (hypoinflammation), seem to be qualified for the use of IgGAM. As a consequence of sepsis-related hyperinflammation, all organs can be affected and numerous biomarkers can be influenced, including CRP, procalcitonin (PCT) and IL-6 [106]. To date there is no clear predictor for the use of IgGAM. The shock index and the qSOFA score (quick sepsis-related organ failure) are also readily available and cost-effective tools for the initial assessment of high-risk patients [107,108].

The measurement of circulating immunoglobulins as biomarkers in sepsis has increasingly become the focus of research within the last ten years. Different approaches are pursued with regard to the development of immune scores as a tool for predicting outcome. A meta-analysis could show that the prevalence of IgG hypogammaglobulinaemia in heterogeneous sepsis cohorts at the time of sepsis diagnosis was as high as 70%, however, a single subnormal measurement of IgG on the day of sepsis diagnosis could not help to identify patients with a higher risk of death [109]. In another study, the combination of several Ig parameters was found to be significantly associated with an unfavourable outcome when levels of IgA, IgG and IgM were below specified cut offs [110]. This finding is consistent with the evidence from a retrospective analysis of 129 patients with septic shock treated with IgGAM, where a survival benefit was associated with the start of treatment within the first 23 h [111].

In addition to the excessive, pro-inflammatory immune response, the syndrome of acquired septic immune paralysis has become the focus of attention, as it seems to be predominantly associated with morbidity and mortality [112]. Although the clinical appearance of hypoinflammation is not as detailed and comprehensive as the phenotype of hyperinflammation, the increased risk of secondary infection in these patients is well known [112]. Patients often develop a chronic critical illness after initial survival of septic shock and cannot leave the intensive care unit. 

Chronic immune paralysis can lead to a renewed increase in the mortality rate after about 30 days; patients often suffer from deleterious nosocomial infections: in addition to the occurrence of multi-resistant bacteria (MDR), viral infections (de novo or reactivations) or infections with fungi are also observed, the successful treatment of which remains one of the major challenges of modern intensive care medicine [113,114,115,116,117]. Possibly, the detection of a low HLA-DR expression on monocytes could serve as an indication of (continued) immune dysfunction and thus indicate increased mortality [118,119,120].

In addition to the identification and selection of suitable patient groups for therapy with IgGAM (“Who?”), the questions of “When?” and “How much?” remain open. There are no clear answers for either aspect. In general, it could be shown that an early start of IgGAM therapy (i.e., 36 h vs. 66 h, respectively) in the presence of an infection leads to a higher probability of survival, which makes the time factor a key influencing variable [121,122].

There is also a lack of evidence on the question of optimal dosage, which is the basis of current studies. In addition to the pros and cons of an initial bolus application, the type of administration (continuous vs. intermittent), the quantity (determined by serum measurements of Ig or without adaptation), but also that of the target value to be achieved (“normal” or “supranormal”) are not sufficiently clarified.

## 5. A Novel Situation in COVID-19

In 2019 the “severe acute respiratory Syndrome coronavirus-2” (SARS-CoV-2) caused a pandemic with an unprecedented global crisis, affecting healthcare and research systems, but also infrastructure sectors, including education, politics and economy. 

According to current knowledge, there is a link between severity of coronavirus disease 2019 (COVID-19), viral production and the severe dysregulation of the inflammatory immune reaction (“cytokine storm”). However, up to date it is still unclear which molecular mechanisms trigger the onset of the immune dysbalance and why it can rapidly progress to multiorgan dysfunction or ARDS with a fatal outcome in a considerable subset of patients [123,124].

Clinical observation of fatal courses of COVID-19 often includes severe acute respiratory distress syndrome (ARDS), which is caused by alveolar injury, and multiple organ failure—both of which are associated with hyperproduction of cytokines [123,124,125]. Both mild and severe/fatal cases display changes in cytokine production, particularly IL-1β, IL-1ra, IL-6, IL-10, TNF, GM-CSF, IL-17, and pathological shifts of circulating leukocyte subsets [126,127].

As a consequence, this leads to the disturbed development of protective immunity against the infection. The most severe complications of COVID-19 include sepsis-like inflammation, pulmonary or cardiovascular complications, and coagulopathy [128,129,130].

As discussed above, the innate immune system of the host is activated in response to the virus to limit infection. Subsequently, the adaptive immune system develops specific immunoglobulins and activates T cells in direct response to the virus.

However, if this inflammation is unmodulated or excessive, there is a risk of chronic hyperinflammation resulting in functional inhibition of the adaptive immune system. In addition to virus-induced lymphopenia, this can result in progressive tissue and organ damage and the failure of the adaptive immune system to develop functional immunoglobulins, thereby clearing the virus [131]. Therefore, the use of IgGAM in patients showing signs of both hyper- and hypoinflammation could be an effective therapeutic strategy.

## 6. Conclusions

To this day, the cornerstones of sepsis and septic shock therapy still consist of timely focus control, the administration of anti-infective drugs and haemodynamic stabilisation through early and sufficient administration of fluid and vasopressors. The clinical understanding of sepsis is evolving towards an immunological perspective and remarkable progress has been made in recent decades. Shedding light on the complex pro- and anti-inflammatory pathways and the disorders of the complementary and coagulation systems, thereby demonstrating the complexity and heterogeneity of the syndrome, has yet not led to transfer this knowledge into evidence-based approaches to treat sepsis and develop effective therapies. Despite progress in the (further) development of innovative therapeutic approaches, such as methods for extracorporeal blood purification, the use of novel anti-infective substances or targeted immune modulation, there are still no therapeutic measures backed up by sufficient evidence that lead to a convincingly reduced mortality in the therapy of sepsis [132].

Therapy with IVIg is a treatment option that is associated with relatively high (but internationally disparate) costs and variable availability of treatment. A targeted selection process of potentially profiting patients on evidence-based criteria in terms of personalized medicine is preferable to an indiscriminate approach. Unfortunately, a reliable and validated assessment of the cost-effectiveness in relation to the total costs of therapy based on available data currently remains difficult to obtain.

Since the approach of using purely anti-inflammatory therapies has been disappointing, the investigation of strategies aiming at balancing the immune system appears to be more appropriate [133]. Given their multiple effects on inflammatory and immune mechanisms, the use of polyclonal intravenous immunoglobulins seems to be a promising approach to both modulate pro- and anti-inflammatory pathways. However, further clinical studies and research are clearly needed to substantiate the rationale for the use of immunoglobulins presented here, and to target the interventions to the right patient subset, at the right time, at the appropriate dose and for an optimal duration.

Summary:The use of IVIg in sepsis and septic shock appears to be safe.The use of IgGAM shows beneficial effects on sepsis-related inflammation and coagulopathy.For adjunctive sepsis therapy with IVIg, IgM-enriched formulations may be advantageous for specific patients.Further clinical data are urgently needed to be able to make definitive statements on the cost-benefit ratio.

## Figures and Tables

**Figure 1 ijms-21-05543-f001:**
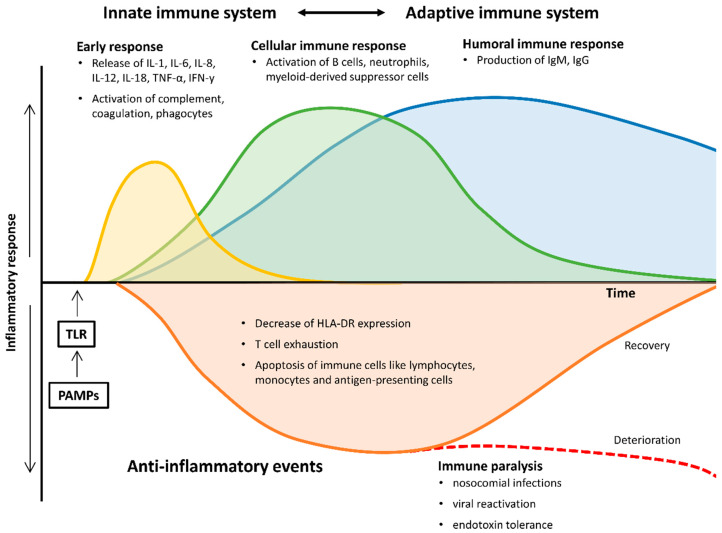
Pro- and anti-inflammatory changes of the immune system during the course of sepsis and septic shock. HLA-DR, human leukocyte antigen-D related; IgM/G, immunoglobulin M/G; IL, interleukin; IFN-γ, Interferon gamma; PAMPs, pathogen-associated molecular patterns; TNF-α, tumor necrosis factor alpha; TLR, toll-like receptor.

**Figure 2 ijms-21-05543-f002:**
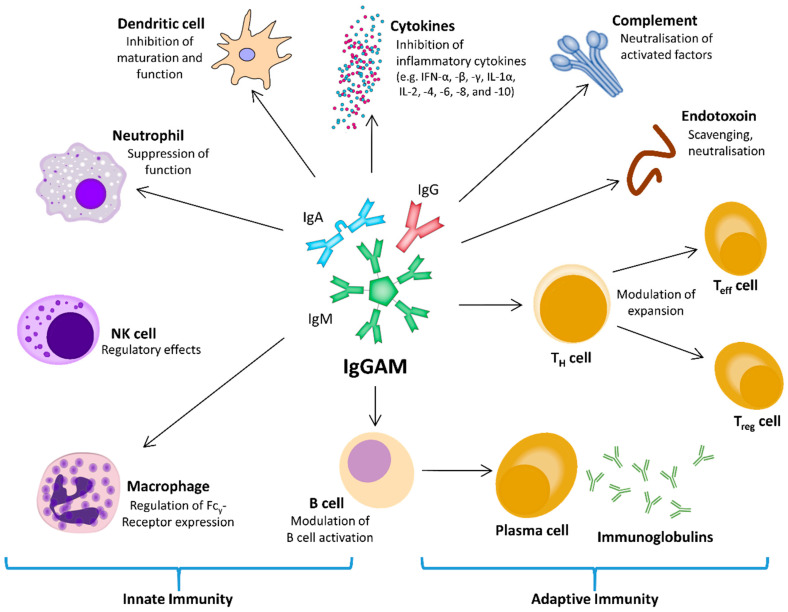
The central role of intravenous immunoglobulins IgGAM on the innate and adaptive immune response, using different regulatory pathways to interact with the cellular and humoral components. IFN, interferon; Ig, immunoglobulin; IgGAM, immunoglobulin G/A/M; IL, interleukin; NK cell, natural killer cell; T_eff_ cell, effector T cell; T_H_ cell, helper T cell; T_reg_ cell, regulatory T cell.

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
