# Peer review of "Use of Intravenous Immunoglobulins in Sepsis Therapy—A Clinical View"

_ijms, 2020, doi:10.3390/ijms21155543_

Round 1

Reviewer 1 Report

This is a well-written manuscript providing a concise description of the immune response in sepsis and the possible role of immunoglobulin therapy in clinical practice. I would appreciate if you could address the following comments:

1. A figure depicting the immune response over time (as described in "2. The changing immune system in sepsis") would help to visualize the role of the immune system. The changes and interaction of pro- and anti-inflammatory  mechanisms over time may also help identify an optimal timepoint to administer various immune therapies.

2. Despite its benefits in theory and animal models, IVIG therapy has not shown clear benefit in clinical studies to date. Could you please expand on whether there are possible risks and adverse events of using immunotherapy in sepsis? For example, could there be a risk of stimulating an already overactive immune system and cause more harm?

3. The cost of therapies, particularly novel and expensive ones such as immunotherapy, plays an increasingly important role in the practice of medicine. Given the large patient population that could potentially benefit from immunotherapy in sepsis, would you be able to provide some range of cost for immunotherapy, and discuss the challenge this may present for healthcare systems?

Author Response

Response to reviewer no. 1:

We thank the reviewer for the important comments.

“1. A figure depicting the immune response over time (as described in "2. The changing immune system in sepsis") would help to visualize the role of the immune system. The changes and interaction of pro- and anti-inflammatory  mechanisms over time may also help identify an optimal timepoint to administer various immune therapies.”

We have added a figure (Figure 1 in the revised manuscript) in which we illustrate the immunologic changes that occur over the time course of sepsis, helping to visualize the simultaneous and partly contrary events.

“2. Despite its benefits in theory and animal models, IVIG therapy has not shown clear benefit in clinical studies to date. Could you please expand on whether there are possible risks and adverse events of using immunotherapy in sepsis? For example, could there be a risk of stimulating an already overactive immune system and cause more harm?”

We have expanded the manuscript (4. "Immunoglobulins in clinical use") to address potential possible risks and adverse events associated with the use of IVIg.

“3. The cost of therapies, particularly novel and expensive ones such as immunotherapy, plays an increasingly important role in the practice of medicine. Given the large patient population that could potentially benefit from immunotherapy in sepsis, would you be able to provide some range of cost for immunotherapy, and discuss the challenge this may present for healthcare systems?”

Addressing the important issue on costs of a therapy using immunoglobulins in sepsis in relation to the clinical benefit, we have also added two insertions under point 4 and the conclusions. However, this question cannot be answered conclusively due to the lack of up-to-date data.

Reviewer 2 Report

It is important issue in sepsis, and it is well reviewed.

I have one quick comment.

As authors reviewed, therapeutic use of IVIGs could be beneficial in selected patients. This could mean IVIGs might be harmful in other selected patients. For example, they cited CIGMA study(ref 72) to demonstrated a beneficial effect of IVIGs in selected patients. Patients with low IgM level had significantly higher survival rate with IVIGs. However, when we calculate, patients without low IgM level have tendency to have higher mortality rate with IVIGs (10/25 vs 5/24).

I recommend  this comment should be included in this review, so the readers would not be misled that IVIGs might no harmful effects.

Author Response

Response to reviewer no. 2:

We thank the reviewer for the important comment.

“As authors reviewed, therapeutic use of IVIGs could be beneficial in selected patients. This could mean IVIGs might be harmful in other selected patients. For example, they cited CIGMA study(ref 72) to demonstrated a beneficial effect of IVIGs in selected patients. Patients with low IgM level had significantly higher survival rate with IVIGs. However, when we calculate, patients without low IgM level have tendency to have higher mortality rate with IVIGs (10/25 vs 5/24).

I recommend  this comment should be included in this review, so the readers would not be misled that IVIGs might no harmful effects.”

We have addressed potential harmful effects of IVIG therapy in the manuscript (4. Immunoglobulins in clinical use). However, we would not conclude that IVIGs lead to excess mortality since the post-hoc analysis of the CIGMA trial did not show a statistically significant difference between patients with IgM levels above 0.8 g/L who received IVIG compared to placebo. However, we have addressed this issue and added a comment as recommended.